# The Psychological Responses of Nurses Caring for COVID-19 Patients: A Q Methodological Approach

**DOI:** 10.3390/ijerph18073605

**Published:** 2021-03-31

**Authors:** Kyung Hyeon Cho, Boyoung Kim

**Affiliations:** 1Balk Medical Center, College of Nursing, Gyeongsang National University, Geoje-si 53290, Korea; zzonedi@naver.com; 2Institute of Health Science, College of Nursing, Gyeongsang National University, Jinju 52727, Korea

**Keywords:** COVID-19, psychological response, nurse, Q methodology

## Abstract

Anxiety among nurses attending patients at the forefront of the global coronavirus disease-19 (COVID-19) pandemic is on the rise. Accordingly, examining the psychological response of nurses who have attended COVID-19 patients is necessary. This study aimed to categorize and characterize the psychological responses of nurses who have experienced COVID-19 patient care based on the Q methodology. The Q-sample extracted 34 statements, and the P-sample marked 22 nurses who had operated a screening clinic and experienced patient care in the local base hospitals treating COVID-19 using convenience sampling. The findings suggest that nurses who have experienced COVID-19 patient care fall into three categories based on the following psychological responses: (1) fear of social stigma, (2) anxiety about the risk of infection, and (3) burden of infection prevention and control nursing. Therefore, it is expected that the results of this study may serve as the basis for emotional support programs that are capable of understanding and mediating the psychological state of nurses who care for patients with COVID-19.

## 1. Introduction

Over the past year, the world has experienced coronavirus disease-19 (COVID-19), an acute respiratory syndrome. The World Health Organization declared COVID-19 a pandemic, which is the highest level of public health emergency of international concern, to block the spread of the disease [1]. COVID-19 is an infectious disease that affects human life and poses a major threat to public health worldwide [2,3].

In the event of a public health crisis caused by infectious diseases, such as COVID-19, healthcare workers have the duty of providing professional care and must take care of patients at their own risk [4]. In particular, nurses in close contact with suspected and confirmed COVID-19 cases provide care while facing personal risks and carrying emotional burdens [5]. During the current pandemic, nurses have been burdened with increasing workloads, from preparing isolation wards to training staff. Nurses are also required to prevent infections in adverse conditions due to the lack of masks and protective equipment [6]. Under these circumstances, nurses are becoming physically and mentally exhausted because they have to perform direct nursing activities, such as contacting confirmed cases, collecting samples, and intravenous therapy, in addition to other duties, such as serving food, cleaning, and disposing of waste [7].

A study in China investigated the psychological effects on 4692 nurses working at government-designated hospitals during the COVID-19 outbreak and found that the mental health of frontline nurses was generally poor because of somatic symptoms (42.7%), depression (9.4%), anxiety (8.1%), and suicidal ideation (6.5%) [8]. Kwon et al. also reported that nurses showed more severe psychiatric symptoms and lower sleep quality than doctors, indicating the serious physical and psychological problems experienced by frontline nurses [9].

Psychological responses are related to vulnerabilities that affect the relevance of physical, psychological, and social resources to individual needs, and are effected by health conditions, motivation, the severity of stressors, various ages, coping strategies, and support systems [10]. Nurses caring for patients with COVID-19 experience extreme psychological reactions, which may cause stress disorders in the long term [11]. Although there are various prior studies related to COVID-19, there is a lack of research on the psychological problems of nurses who have cared for COVID-19 patients that reflect the circumstances at the time of the outbreak [9]. Research needs to be performed to objectively classify the subjectivity of psychological responses of nurses who have experienced caring for COVID-19 cases [12]. These responses can be very personal and subjective, so it is necessary to understand the underlying essence and interpret these by applying the Q methodology [13]. Q methodology uses statements that describe people’s thoughts, attitudes, and values about an object or topic to examine similarities and correlations between people; this method enables researchers to sort similar people into several distinct types and to discover common factors [14]. The meaning of each statement in Q methodology is empirical. Therefore, a large sample is unnecessary because it is not about analyzing categorical aggregates but rather looking at the differences between factors (which may be factors created by one person) [15].

Therefore, this study aimed to prepare basic data for the psychological support of frontline nurses by classifying and analyzing the psychological responses of nurses who have experienced caring for COVID-19 patients through Q methodology.

## 2. Materials and Methods

### 2.1. Research Process and Procedure

This is an exploratory study applying the Q methodology to the psychological responses of nurses who have cared for COVID-19 patients. Data were collected from 17 September 2020 to 15 October 2020, and the study was conducted in six stages. The first phase involved building the Q-population (concourse) through a literature review and in-depth interviews using semi-structured questions. Phase 2 involved selecting the Q-sample from the concourse, creating cards, and preparing a distribution map of the Q-sample. The P-sample was selected in Phase 3, and Q-sorting and post-interviews were conducted in Phase 4. Q-sorting was performed according to a forced quasi-normal distribution with a scale labeled “most agree,” “neutral,” and “most disagree.” The data were processed and analyzed in Phase 5, and the final phase involved interpreting the results (Figure 1).

### 2.2. Building the Q-Population (Concourse)

#### 2.2.1. Q-Population

This study extracted the Q-population through a literature review and in-depth interviews using semi-structured questions. The interviews were conducted with six nurses who had experienced caring for COVID-19 patients. Each interview took approximately 30 to 40 min. The semi-structured interview questions were: “What did you experience while caring for COVID-19 patients?”; “What were the psychological challenges in caring for COVID-19 patients?”; and “How do you think your experience of nursing COVID-19 patients has affected your life as a nurse?”. The researcher recorded and transcribed the interviews after obtaining informed consent from the participants and extracted 100 statements related to the psychological responses of nurses through the interviews.

A total of 225 statements related to the psychological responses of nurses were extracted by reviewing the following literature: “The Emotional Distress and Fear of Contagion Related to Middle East Respiratory Syndrome (MERS) on General Public in Korea” [16], “Nurses’ Experience of Middle East Respiratory Syndrome Patients Care” [17], “Turnover Intention of Nurses that were Cohort Quarantined During the Middle East Respiratory Syndrome (MERS) Outbreak” [18], “Psychological Reactions and Physical Trauma by Types of Disasters” [19], “The validation of Korean Version of the Psychological Well Being-Post Traumatic Changes” [20], and the “Guidelines for Psychological Support Related to COVID-19” [21,22].

#### 2.2.2. Q-Sampling

To select the Q-sample, the researcher reviewed a total of 325 Q-population statements (100 through interviews + 225 through literature review) several times, eliminated repetition, and categorized those with similar values or meanings. During this process, 160 statements were extracted, focusing on their relevance to psychological reactions, after consulting with two professors specializing in qualitative research. Subsequently, 60 preliminary statements were extracted from the 160 statements. Then, 40 statements were extracted by reviewing the statements with two nurses who had experience caring for COVID-19 patients. The content validity index of the 40 statements was evaluated by three professors in qualitative research and two nurses who had experience caring for COVID-19 patients. As a result, 34 final statements with a content validity index of 0.8 or higher were chosen as the Q-sample [15,23,24].

#### 2.2.3. Reliability Test

As a result of evaluating the internal consistency by sorting the final 34 selected statements with three participants to verify the reliability, the reliability was high (r = 0.94).

#### 2.2.4. P-Sample

Based on the study by Kim [15], the size of the P-sample should be 20 ± 10 people. Thus, 22 nurses who worked at general hospitals in A city and Y city, North Gyeongsang Province, and G city, South Gyeongsang Province, and had experience working in screening clinics or nursing suspected or confirmed COVID-19 cases were selected as the P-sample. The subjects consisted of 22 nurses who had experienced caring for patients at COVID-19 screening clinics or regional hospitals. Their average age was 38 years (mean (standard deviation [SD]) = 38.91 (8.87)), and their clinical experience varied from 8 months to 30 years, but most were experienced nurses, with 16 of them having more than 10 years of clinical experience. The subjects worked in the intensive care unit (ICU) (6), general ward (3), emergency room (ER) (3), and outpatient departments (10). Their current positions were as follows: 13 general nurses, 1 charge nurse, 6 head nurses, 1 nursing supervisor, and 1 assistant manager of nursing (Table 1).

#### 2.2.5. Q-Sorting

To collect data, 34 statement cards made from the Q-sample were given to the P-sample who agreed to participate in the study. To sort the Q-sample effectively and efficiently, the participants read all the statements and primarily sorted them into 11 “agree,” 11 “disagree,” and 12 neutral statements according to their subjective opinion. After this process, the participants ranked the Q-statements by choosing two statements that were “most agree” and gave them +4 points, then chose the next three statements and gave them +3 points, then chose the next four statements and gave them +2 points, and moved forward by arranging the rest of the statements from the outside right to the inside. The 11 “most disagree” statements were ranked in the same way starting from the –4 slots on the outside left, and the statements that were difficult to sort were arranged as neutral. After positioning the cards, the participants were reminded that the positioning of the statements could be changed or moved at any time during sorting. During the post-interviews, the participants were asked why they chose and positioned the statements in their respective ways.

### 2.3. Data Processing and Analysis Method

After completing the sorting process with the selected 22 participants, the statements were ranked in the order of 1 point (−4), 2 points (−3), 3 points (−2), 4 points (−1), 5 points (0), 6 points (+1), 7 points (+2), 8 points (+3), and 9 points (+4) starting from the “most disagree” statements. In terms of data analysis, the types were classified by principal component factor analysis using the PC QUANL program, and the characteristics of each type were analyzed based on the difference between the standard score for each type and the standard score between each type, the difference between the standard score and the average standard score for each type, and the matching items and their standard score results.

### 2.4. Ethical Considerations

In terms of ethical considerations, this study was approved by the Institutional Review Board of G University through a deliberation process (GIRB-G20-Y-0041). The purpose of the study, the interview process, and requirements for the interview were explained to the subjects before collecting the data. The research was conducted after obtaining written consent, and the participants were informed that they may withdraw their consent at any time and discontinue participation without penalty. The consent form also guaranteed anonymity and confidentiality. The form also explained that the responses would only be used for research and that the data and consent would be stored for three years after study completion, at which point the data would be confidentially discarded or deleted.

## 3. Results

### 3.1. Forming the Q-Types

As a result of analyzing the Q-factors for the psychological responses of nurses who had experience caring for COVID-19 patients, 34 statements were classified into three types (Table 2). The three types accounted for 56% of the total: type 1 (39%, *n* = 12), type 2 (10%, *n* = 7), and type 3 (7%, *n* = 3). The correlation between the types was distributed from r = 0.10 to 0.44. The lowest was between type 1 and type 3 (r = 0.10), and the highest was between type 1 and type 2 (r = 0.44) (Table 3).

### 3.2. The Types of Psychological Responses from Nurses with Experience Caring for COVID-19 Patients

The psychological responses from nurses who experienced caring for COVID-19 patients were categorized into three types: “fear of social stigma,” “anxiety about the risk of infection,” and “burden of infection prevention and control nursing.”

#### 3.2.1. Type 1: Fear of Social Stigma

There were 12 type 1 participants, and their most agreed with statements were Q3, Q5, Q2, Q4, and Q14 (Table 4). On the other hand, their most disagreed with statements were Q33, Q21, Q30 Q31, Q32, Q34, and Q19. Compared to the other types, type 1 agreed more with Q-statements 5, 14, 1, and 8 (z-score difference >1), and disagreed more with Q-statements 34, 30, 24, 28, and 31 (z-score difference <−1) (Table 5).

Participant 12 had a type 1 weighting factor of 4.42, and she was a head nurse with 24 years of clinical experience who had experience working at screening clinics and caring for suspected and confirmed cases. Her most agreed with statements were Q3 and Q5. She said, “I was worried that I might spread COVID-19 to my family, so I did not go home and stayed at the hospital, and felt sorry about taking care of the patients before my family. I was also upset and sorry because people were wary of my family because of me, even though they were not infected.”

The participants belonging to type 1 were characterized as being very upset, because they were wary of stigmatizing their families due to their work environment, even though their families were not confirmed cases. They also feared that they might spread COVID-19 to their families because they were exposed to the risk of infection, and that their families might unintentionally spread COVID-19 and become a target of social stigma.

#### 3.2.2. Type 2: Anxiety about the Risk of Infection

In type 2, the most agreed with Q-statements were Q3, Q24, Q2, Q4, and Q1, and the most disagreed with Q-statements were Q34, Q22, Q26, and Q9, respectively. Compared to the other types, type 2 agreed more with Q-statements Q2, Q30, Q3, Q34, and Q24, and disagreed more with Q-statements Q9, Q6, Q7, Q12, and Q13.

Participant 1 had the highest factor weight (1.95) among those classified as type 2, and she was a head nurse with 28 years and 7 months of clinical experience who had experience in working at screening clinics and caring for suspected and confirmed cases. She said, “I was anxious about contracting COVID-19 because I am a head nurse working in the intensive care unit, so if I get infected, I might spread the virus to critical patients. I am worried that other people might contract the virus because of my work environment. People become more sensitive when healthcare professionals get infected, and I am worried that patients with weak immune systems will get infected by me.”

Participants in type 2 were characterized as being extremely alert about the risk of infection, feeling anxiety and tension that they might be infected due to their carelessness, bearing the physical pain caused by the difficulty of wearing goggles or protective equipment, and valuing the importance of protective measures.

#### 3.2.3. Type 3: Burden of Infection Prevention and Control Nursing

In type 3, the most agreed with Q-statements were Q11, Q4, 24, Q12, and Q28, and the most disagreed with Q-statements were Q5, Q2, Q25, and Q21. Compared to the other types, type 3 agreed more with Q-statements Q11, Q28, Q7, and Q6, and disagreed more with Q-statements Q2, Q5, and Q3.

Participant 5 had a factor weight of 1.13 among those classified as Type 3, and she was a general nurse with 3 years and 7 months of clinical experience who had no experience in working at screening clinics but cared for suspected and confirmed cases. She said, “Nursing care itself is difficult and exhausting, but when I have to care for patients while wearing protective clothing and equipment, I feel more tired, hot, and humid, and I feel uncomfortable and irritated more easily. COVID-19 seems like an endless fight, so this makes my body and mind more tired.” On the other hand, she also said, “The products were certified by the state, so I believe that wearing masks will prevent most viruses. I think that KF94 masks are safe, and I have no doubt about N95 masks.”

Participants belonging to type 3 were characterized by strong belief in K-quarantine, and fatigue and boredom from nursing infectious diseases, such as quarantine measures for infection control and difficulties in wearing protective equipment, rather than being anxious due to the prolonged COVID-19 pandemic.

## 4. Discussion

The purpose of this study was to investigate the types of psychological responses and characteristics of nurses caring for COVID-19 patients to pave the way for emotional support for nurses fighting COVID-19.

The COVID-19 nurses who experienced caring for COVID-19 patients showed three types of psychological responses: “fear of social stigma,” “anxiety about the risk of infection,” and “burden of infection prevention and control nursing.” The statement with a high z-score in types 1 and 2 was, “I am worried that COVID-19 might spread in my family because of my work environment,” showing anxiety about infection, with a correlation of 0.44. However, the difference between these two types was that type 1 participants were extremely afraid of the social stigma that their families might be treated as super-spreaders because of them, and type 2 participants were more anxious about contracting COVID-19 and becoming confirmed cases.

According to the results of this study, all three types agreed with the Q-statement of “I am more afraid of what other people will think if I contract COVID-19, as a nurse.” According to the P-sample interviews who chose this statement, “People get more sensitive when a nurse contracts COVID-19 than when an ordinary person becomes a confirmed case. In particular, they might think that we can become super-spreaders to patients with weak immune systems in hospitals. Sometimes, we feel guilty because people have high expectations about our cleanliness and medical knowledge about infectious diseases. There is also a huge psychological burden because of the possibility of spreading COVID-19 to other people.” The most difficult psychological burden of nurses, who are more exposed to the risk of infection than ordinary people, was found to be the pressure of not contracting COVID-19.

Specifically, type 1 was characterized by “fear of social stigma.” Due to concerns about infecting their families during the current pandemic and their job as nurses caring for COVID-19 patients, they had psychological wounds because people might brand their families as COVID-19 spreaders. A study by Nie et al. [3] on the psychological impact of the COVID-19 outbreak on frontline nurses also reported concerns about their families, being treated differently, and psychological distress about negative coping styles, which was consistent with the results of this study. During the COVID-19 pandemic, nurses in Australia also expressed concern about work-related risks to themselves and their families, in addition to common psychological reactions such as stigma and anxiety about infection [25]. A study in Nepal reported that healthcare workers facing stigma were more at risk of developing exhaustion, fatigue, and psychological distress, which affected their work concentration [26]. Similar to type 1, type 2 frontline nurses also experienced psychological distress because of the prejudice of close neighbors viewing them as COVID-19 spreaders rather than recognizing and respecting them as nurses caring for patients.

Type 2 was characterized by anxiety about the risk of infection, showing a high level of alertness about the environment and anxiety about infection. Although they worried about contracting COVID-19, they did not complain much about the fatigue or physical pain caused by wearing protective equipment, because of their strong belief that such equipment can prevent infections. Type 2 participants strongly agreed with the Q-statement of “I am worried that COVID-19 might spread in my family because of my work environment.” However, unlike type 1, they were more anxious about the risk of their families contracting COVID-19 than their fear of social stigma. Frontline nurses during the Middle East respiratory syndrome (MERS) outbreak were anxious about the risk of transmission and expressed concern about infecting their colleagues and families because of their contact with MERS patients [17]. A study in the Philippines reported that 37.8% of nurses experienced COVID-19 anxiety during the pandemic [27]. A study by Wu et al. [28] reported that nurses showed high anxiety about infection due to poor sleep quality and the possibility of infecting themselves and their families, and suggested the need to intervene with strategies to help physical and mental health. Type 2 participants expressed fear of infection, and the excessive stress caused by such anxiety, tension, and reduced confidence may have a detrimental effect on the mental health of nurses.

Type 3 was characterized by the “burden of infection prevention and control nursing”, showing lower levels of anxiety than the other types, and emphasized the importance of protective measures to overcome this crisis. They were less anxious because of their strong belief in K-quarantine and the masks and protective equipment certified by the state. Type 3 participants faithfully performed their duties as nurses during the pandemic and were burdened because of fatigue and boredom, due to difficulties in wearing protective equipment and complying with quarantine measures for infection control, rather than experiencing anxiety due to the prolonged COVID-19 pandemic. They dealt with the stress of nursing activities and experienced physical and psychological difficulties in wearing protective clothing while caring for COVID-19 patients quickly and accurately. During infectious disease outbreaks, healthcare workers are exposed to various risk factors, which may cause safety problems and severe stress [29].

In general, nurses are stressed out because of heavy workloads and are always exposed to risk due to various infectious diseases, biological hazards, carcinogens, demanding psychological and physical labor, and shift work [30]. Therefore, the outbreak of infectious diseases will inevitably add more stress to nursing work. We will have to manage and resolve the work stress of nurses because the burden of managing patients will accumulate during the current pandemic and increase their already heavy workloads. Therefore, effective measures need to be developed to raise the morale of nurses fighting COVID-19, which is an important strategy to deal with the current pandemic.

This study investigated the psychological responses of nurses caring for COVID-19 patients. The study found that nurses need emotional support to detect negative psychological responses at an early stage to prevent post-traumatic stress disorder. Evidence-based infectious disease training for COVID-19 is a technical support to eliminate risk factors for new infectious diseases [31]. Such technical support can act as a point of pride and subjective reward for frontline nurses fighting against COVID-19, which could offset the fear of stigma. Furthermore, it can help nurses psychologically accept and cope with their fear of infection. Of course, personal anxiety and fears about contracting the virus may not be resolved or improved simply by fulfilling their social responsibilities as nurses caring for COVID-19 patients. However, if members of society sincerely sympathize with nurses and give them positive evaluations, nurses can reduce their negative emotions and psychological reactions and receive comfort and support. Therefore, members of the public should create an atmosphere of support for nurses to alleviate their fear of infection.

The Korean government is operating a national trauma center to provide psychological support for trauma patients to help cope with stress and mental health during the COVID-19 pandemic [32]. The nurses caring for confirmed cases experience fatigue due to their heavy workloads from three rotating shifts and the heavy protective equipment they are required to wear [33]. It is difficult for nurses to receive mental health services in such an environment. Therefore, a support system should be established at an institutional level so that nurses can receive online or telephone counseling services for psychological support without worrying about what other people may think of them and with no environmental constraints.

This study is meaningful in providing useful information on psychological support for nurses by examining the responses of nurses who cared for suspected and confirmed COVID-19 cases and identifying their psychological characteristics. Nevertheless, this study has some limitations. First, the participants were nurses in Korea; therefore, there are limits to generalizing the results. Second, this study focuses on subjectivity, so it cannot represent the types of psychological responses of nurses in Korea.

The following are suggestions made for future research. Similar to the three psychological types of nurses, quantitative studies should be conducted to explore the relationship between various variables, such as social stigma, fear, anxiety, and fatigue. This study only reflects the current COVID-19 pandemic, so further research should be performed on the psychological impact and trauma of nurses after this pandemic ends. The experience of taking care of COVID-19 patients will lead to both negative and positive responses. Therefore, follow-up studies should also be performed on the post-traumatic growth of nurses based on positive experiences during crisis situations, such as infectious diseases.

## 5. Conclusions

Classifying the psychological responses of nurses who experienced caring for COVID-19 patients allowed us to investigate the emotional difficulties they face as well as the values they hold. Nurses caring for COVID-19 patients in Korea are experiencing fear of social stigma, anxiety about infection, and fatigue from heavy workloads. Their most difficult psychological burden turned out to be the pressure and fear of contracting COVID-19. They were also more afraid that their families would contract the virus than themselves. This result reflects the characteristics of Korean culture, which values the importance of family. Although invisible, victims may suffer from psychological distress for a long time. Therefore, we hope that the findings of this study will help to shed light on the psychological reactions and pain of nurses caring for COVID-19 patients, and we believe they can be used as basic data for short and long-term psychological support programs.

## Figures and Tables

**Figure 1 ijerph-18-03605-f001:**
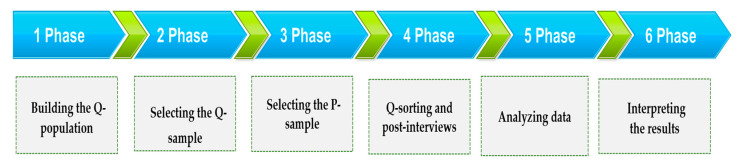
Research process and procedure.

**Table 1 ijerph-18-03605-t001:** General characteristics and factor weights of the P-sample for each type (*n* = 22).

Type	P-Sample	Factor Weight	Gender	Age	Marital Status	Clinical Experience	Current Position	Clinic Work
Type 1 (*n* = 12)	p4	0.45	F	32	Married	11 years, 10 months	General nurse	ICU
	p6	0.43	F	23	Single	8 months	General nurse	General ward
	p8	1.28	F	38	Married	15 years, 8 months	General nurse	ICU
	p9	0.90	F	46	Married	23 years, 6 months	Head nurse	Outpatient department
	p10	1.00	F	38	Married	15 years	General nurse	Outpatient department
	p11	1.35	F	42	Married	17 years, 2 months	Charge nurse	General ward
	p12 *	4.42	F	48	Married	24 years	Head nurse	Outpatient department
	p13	3.43	F	36	Married	9 years, 9 months	General nurse	Outpatient department
	p14	3.10	F	41	Married	15 years, 10 months	General nurse	Outpatient department
	p15	2.06	F	40	Married	12 years, 4 months	General nurse	Outpatient department
	p16	3.19	F	40	Married	18 years	Head nurse	ER
	p22	1.04	F	37	Married	10 years	General nurse	Outpatient department
Type 2 (*n* = 7)	p1 *	1.95	F	50	Married	28 years, 7 months	Head nurse	ICU
	p2	0.83	F	49	Married	20 years	General nurse	Outpatient department
	p7	0.91	F	28	Single	6 years, 6 months	General nurse	Outpatient department
	p17	0.94	F	44	Married	22 years	Other	ER
	p18	1.07	F	44	Married	18 years	Head nurse	ER
	p19	0.92	F	35	Married	8 years, 2 months	General nurse	General ward
	p21	1.05	F	35	Married	13 years, 8 months	Head nurse	ICU
Type 3 (*n* = 3)	p3	0.66	F	26	Single	3 years, 10 months	General nurse	ICU
	p5 *	1.13	F	25	Single	3 years, 7 months	General nurse	ICU
	p20	0.86	F	59	Married	30 years	Other	Outpatient department

* Typical examples of each type. ICU: intensive care unit; ER: emergency room.

**Table 2 ijerph-18-03605-t002:** Factor matrix.

No.	Q-Statements	Type 1	Type 2	Type 3
1	I am confused because it is my first time experiencing COVID-19.	1.9	1.1	0.5
2	I am worried that I might contract COVID-19.	1.5	1.8	−2.5
3	I am worried that COVID-19 might spread in my family because of my work environment.	2.0	2.5	−0.2
4	I am more afraid of what other people will think if I contract COVID-19, as a nurse.	1.2	1.8	1.5
5	I feel sorry for my family because I have to work at a COVID-19 screening hospital.	1.6	0.4	−2.5
6	I feel bored because COVID-19 is not going away any time soon.	0.0	−0.9	0.6
7	I am sad because confirmed patients die without feeling the warmth of their family at the end of their lives.	0.1	−0.8	0.7
8	I am always worried about whether the protective suits and N95 masks are really safe.	0.3	−0.8	−0.8
9	I am worried that wearing goggles will leave a scar on my forehead.	−0.4	−1.5	0.0
10	I am always nervous because I might contract the virus while taking off my protective suit and equipment.	0	0.2	0.0
11	I feel irritated because of the moisture and sweat caused by wearing protective suits and goggles.	0.1	−0.3	2.2
12	I feel dizzy due to wearing Level-D protective suits for a long time.	0.9	−0.1	1.2
13	My whole body ached, and I struggled with headaches and muscle pain after taking care of patients while wearing a Level-D protective suit.	0.6	−0.7	0.2
14	I am hurt because my husband and children are treated unfairly because I work at a designated COVID-19 treatment hospital.	1.0	−0.4	−0.2
15	I am burdened by the reality that my work as a nurse comes before the health of my family during the current pandemic.	0.8	0.2	0.8
16	I feel the pressure of protecting myself due to the COVID-19 guidelines.	0.4	0.1	0.7
17	If another type of infectious disease crisis occurs, I have the confidence to deal with it through my past experience.	0.5	−0.8	−0.1
18	I realized that I am stronger than I thought during this crisis.	0.8	−0.8	0.6
19	I try to avoid uncomfortable situations.	−1.1	−0.9	−0.1
20	I would rather take care of general patients instead of COVID-19 patients during my working hours.	−0.8	−0.1	−0.1
21	Currently, I regret becoming a nurse.	−1.5	−0.9	−1.1
22	I think I can never go back to my life before COVID-19.	−0.4	−1.3	−0.9
23	I don’t think anywhere in the world is safe from COVID-19.	−0.7	0.0	0.1
24	I get upset and angry when I see suspected and confirmed cases not following self-quarantine measures.	0.1	2.0	1.4
25	I feel distant or disconnected from the people around me.	−0.5	−0.7	−1.3
26	I don’t have the time or luxury to heal my psychological wounds.	−1	−1.3	−0.9
27	I had a valuable and helpful experience because of the COVID-19 pandemic.	0.0	0.3	0.4
28	I think warm supportive messages give us strength.	−0.4	0.4	1.2
29	My family respects and recognizes my job as a dedicated nurse for COVID-19 patients.	−0.2	1.0	0.4
30	I became overly wary of my environment, fearing that I may have crossed paths with confirmed cases.	−1.4	0.8	−0.2
31	I avoid personal relationships because I am a nurse working at a screening center.	−1.4	−0.1	−0.6
32	I suspect even close acquaintances because of the fear of crossing paths with confirmed cases.	−1.2	0.0	−0.6
33	I have anorexia and indigestion because of my heavy workload of caring for COVID-19 patients.	−1.7	−1.0	−0.9
34	I do my best to care for COVID-19 patients, but my self-esteem decreases when patients demand unfair or inappropriate care.	−1.2	1.1	0.3

**Table 3 ijerph-18-03605-t003:** Correlations and eigenvalues by type.

Type	Type 1	Type 2	Type 3
Type 1	1		
Type 2	0.44	1	
Type 3	0.10	0.14	1
Eigenvalue	8.54	2.27	1.41
Variance (%)	38.82	10.34	6.39
Cumulative	38.82	49.15	55.55

**Table 4 ijerph-18-03605-t004:** Q-statements and array Z-scores for each type and consensus items.

Type	Q-Statements *	Z-Scores
1	3. I am worried that COVID-19 might spread in my family because of my work environment.	2.01
	1. I am confused because it is my first time experiencing COVID-19.	1.91
	5. I feel sorry for my family because I have to work at a COVID-19 screening hospital.	1.58
	2. I am worried that I might contract COVID-19.	1.52
	4. I am more afraid of what other people will think if I contract COVID-19, as a nurse.	1.23
	14. I am hurt because my husband and children are treated unfairly because I work at a designated COVID-19 treatment hospital.	1.01
	19. I try to avoid uncomfortable situations.	−1.08
	34. I do my best to care for COVID-19 patients, but my self-esteem decreases when patients demand unfair or inappropriate care.	−1.17
	32. I suspect even close acquaintances because of the fear of crossing paths with confirmed cases.	−1.19
	31. I avoid personal relationships because I am a nurse working at a screening center.	−1.42
	30. I became overly wary of my environment, fearing that I may have crossed paths with confirmed cases.	−1.42
	21. Currently, I regret becoming a nurse.	−1.46
	33. I have anorexia and indigestion because of my heavy workload of caring for COVID-19 patients.	−1.68
2	3. I am worried that COVID-19 might spread in my family because of my work environment.	2.46
	24. I get upset and angry when I see suspected and confirmed cases not following self-quarantine measures.	1.97
	2. I am worried that I might contract COVID-19.	1.82
	4. I am more afraid of what other people will think if I contract COVID-19, as a nurse.	1.75
	1. I am confused because it is my first time experiencing COVID-19.	1.07
	34. I do my best to care for COVID-19 patients, but my self-esteem decreases when patients demand unfair or inappropriate care.	1.05
	22. I think I can never go back to my life before COVID-19.	−1.30
	26. I don’t have the time or luxury to heal my psychological wounds.	−1.35
	9. I am worried that wearing goggles will leave a scar on my forehead.	−1.53
3	11. I feel irritated because of the moisture and sweat caused by wearing protective suits and goggles.	2.16
	4. I am more afraid of what other people will think if I contract COVID-19, as a nurse.	1.55
	24. I get upset and angry when I see suspected and confirmed cases not following self-quarantine measures.	1.37
	12. I feel dizzy due to wearing Level-D protective suits for a long time.	1.24
	28. I think warm supportive messages give us strength.	1.23
	21. Currently, I regret becoming a nurse.	−1.05
	25. I feel distant or disconnected from the people around me.	−1.30
	2. I am worried that I might contract COVID-19.	−2.47
	5. I feel sorry for my family because I have to work at a COVID-19 screening hospital.	−2.48

* Q statements with Z-scores greater than +1.0 or less than −1.0.

**Table 5 ijerph-18-03605-t005:** Z-Score differences between types.

Type	Q Statements	Z-Score Difference *
1	5. I feel sorry for my family because I have to work at a COVID-19 screening hospital.	2.61
	14. I am hurt because my husband and children are treated unfairly because I work at a designated COVID-19 treatment hospital.	1.32
	1. I am confused because it is my first time experiencing COVID-19.	1.13
	8. I am always worried about whether the protective suits and N95 masks are really safe.	1.05
	31. I avoid personal relationships because I am a nurse working at a screening center.	−1.07
	28. I think warm supportive messages give us strength.	−1.16
	24. I get upset and angry when I see suspected and confirmed cases not following self-quarantine measures.	−1.58
	30. I became overly wary of my environment, fearing that I may have crossed paths with confirmed cases.	−1.73
	34. I do my best to care for COVID-19 patients, but my self-esteem decreases when patients demand unfair or inappropriate care.	−1.85
2	2. I am worried that I might contract COVID-19.	2.30
	30. I became overly wary of my environment, fearing that I may have crossed paths with confirmed cases.	1.61
	3. I am worried that COVID-19 might spread in my family because of my work environment.	1.57
	34. I do my best to care for COVID-19 patients, but my self-esteem decreases when patients demand unfair or inappropriate care.	1.48
	24. I get upset and angry when I see suspected and confirmed cases not following self-quarantine measures.	1.24
	13. My whole body ached, and I struggled with headaches and muscle pain after taking care of patients while wearing a Level-D protective suit.	−1.13
	12. I feel dizzy due to wearing Level-D protective suits for a long time.	−1.16
	7. I am sad because confirmed patients die without feeling the warmth of their family at the end of their lives.	−1.20
	6. I feel bored because COVID-19 is not going away any time soon.	−1.22
	9. I am worried that wearing goggles will leave a scar on my forehead.	−1.34
3	11. I feel irritated because of the moisture and sweat caused by wearing protective suits and goggles.	2.27
	28. I think warm supportive messages give us strength.	1.23
	7. I am sad because confirmed patients die without feeling the warmth of their family at the end of their lives.	1.11
	6. I feel bored because COVID-19 is not going away any time soon.	1.04
	3. I am worried that COVID-19 might spread in my family because of my work environment.	−2.48
	5. I feel sorry for my family because I have to work at a COVID-19 screening hospital.	−3.48
	2. I am worried that I might contract COVID-19.	−4.15

* Comparing with other type, Z-score difference with greater than +1.0 or less than −1.0.

## Data Availability

Data is available upon substantiated request from the corresponding author.

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
