# Peer review of "The Psychological Responses of Nurses Caring for COVID-19 Patients: A Q Methodological Approach"

_ijerph, 2021, doi:10.3390/ijerph18073605_

Round 1
Reviewer 1 Report
This study presents some very interesting results that will help to prevent the development of post-traumatic stress in this cases.
In general it seems to me a good study on the methodology used and the results obtained. However, I have small comments that I hope will help to improve some aspects.
In the materials and methods, it would be interesting to incorporate a figure that contains a summary of the phases that are explained in general in lines 65-74.
The table 1 not read properly at the top: type, P-sample, ... the letters are too close together, maybe it should be improved to make it look better. In addition, table 1 is very far from being mentioned in the text, in line 120.
The conclusions are too short. Perhaps the originality of this study should be explained more in front of others.
Also check some spaces in the references, for example line 340, line 359.
Line 361: the author Jung, H is not good, because appears: “ung, H”.
Also review the reference styles marked by the magazine, it seems that it does not comply with the specified format.
Author Response
Manuscript ID: ijerph-1130226
Type of manuscript: Article
Title: The Psychological Responses of Nurses Caring for COVID-19 Patients: A Q Methodological Approach
Authors: Kyung hyeon Cho, Boyoung Kim *
Dear Editors:
Thank you very much for reviewing our study. We have attempted to address each of the reviewers' comments, and in so doing, we believe we have improved the manuscript substantially. The revisions made to the manuscript are highlighted in red font. We have also included point-by-point responses to each comment.

Reviewer 2 Report
I believe that the introductory section could be improved by increasing its length and focusing on the psychological factors of nurses' experiences.
The Q methodology is still unknown to most of the professionals who might read this document, it might be interesting to explain this methodology in more detail.
Author Response
Manuscript ID: ijerph-1130226
Type of manuscript: Article
Title: The Psychological Responses of Nurses Caring for COVID-19 Patients: A Q Methodological Approach
Authors: Kyung hyeon Cho, Boyoung Kim *
Dear Editors:
Thank you very much for reviewing our study. We have attempted to address each of the reviewers' comments, and in so doing, we believe we have improved the manuscript substantially. The revisions made to the manuscript are highlighted in red font. We have also included point-by-point responses to each comment, which are included below (shown in blue):

Reviewer 3 Report
The study is important and very interesting. However there is some parts which require improvements:
1) I methodological part: there is a need to describe research group (p-sample) more deeply. More socio-demographic information is needed.
2) In section 2.4: it should contain more information about the procedure of collecting data. Was is anonymous?, Did participants have a knowledge about the purpose of the study? Did they give a consent to take part in the study?
3) Tables 3 and 4 are not clear. Which Q statements refer to which type 1,2 and 3?
4) In the discussion part Authors write: "
The purpose of this study was to investigate the types of psychological responses and characteristics of nurses caring for COVID-19 patients to pave the way for emotional support for nurses fighting COVID-19." (line 220-222). However, they didn't write about the ways of emotional support which should be appropriate to each type of psychological responses.
Author Response

(The authors gave the same response as above.)
